# Calculation Method of Permanent Deformation of Asphalt Mixture Based on Interval Number

**DOI:** 10.3390/ma14092116

**Published:** 2021-04-22

**Authors:** Yue Xiao, Limin Tang, Jiawei Xie

**Affiliations:** 1School of Traffic and Transportation Engineering, Changsha University of Science and Technology, Changsha 410004, China; xiao_yue93@163.com; 2Hunan Provincial Communications Planning, Survey & Design Institute, Co., Ltd., Changsha 410008, China; 3College of Materials Science and Engineering, Hunan University, Changsha 410082, China; xjwcwhnu@163.com

**Keywords:** road design theory, asphalt mixture permanent deformation, interval formula, uncertainty, interval analysis theory

## Abstract

There are great uncertainties in road design parameters, and the traditional point numerical calculation results cannot reflect the complexity of the actual project well. Additionally, the calculation method of road design theory based on interval analysis is more difficult in the use of uncertain design parameters. In order to simplify the calculation process of the interval parameters in the road design theory, the asphalt pavement design is taken as the analysis object, and the permanent deformation of the asphalt mixture is simplified by combining the interval analysis theory. Considering the uncertainty of the design parameters, the data with boundaries but uncertain size are expressed in intervals, and then the interval calculation formula for the permanent deformation of the asphalt mixture is derived, and the interval results are obtained. In order to avoid the dependence of interval calculation on the computer code, according to the interval calculation rule, the interval calculation method with the upper and lower end point values as point operations is proposed. In order to overcome the contradiction between interval expansion results and engineering applications, by splitting the multi-interval variable formulas, the interval variable weights are reasonably given, and the synthesis of each single interval result realizes a simplified calculation based on interval variable weight assignment. The analysis results show that the interval calculation method based on the point operation rule is accurate and reliable, and the simplified method based on the interval variable weight assignment is effective and feasible. The simplified interval calculation method proposed in this paper provides a reference for the interval application of road design theory.

## 1. Introduction

In China, long-life and high-performance road structure design theory is a hot research topic [1]. The theoretical analysis method of reliability has an important foundation function for the design of long-life road structures, and the reliability coefficient obtained from experience is widely used in road design. At present, the design theory of highways in China adopts the Mechanistic–Empirical method [2]. When calculating the permanent deformation of asphalt mixture, its influence parameters and calculation models have been studied by many scholars [3,4]. However, the reliability coefficient of this fixed value does not reflect the essential situation of a specific design project, and especially in the requirements of new pavement structure design, it is urgent to propose a more reliable analysis method. With the deepening of cognition, reliability analysis is considered to be an effective method to explain the uncertainty of design parameters [5]. Additionally, the importance of reliable estimates of travel demand for effective planning, design, and management of roads and facilities is well known by transportation engineers. Therefore, obtaining a higher accuracy of annual average daily traffic (AADT) volume is a challenging task. However, due to the inevitable errors of statistical tools and methods, the use of AADT point values is unreliable [6,7,8]. In addition, measurement uncertainty is considered to be widespread in the field of road engineering [9,10,11,12,13]. According to the theory of measurement uncertainty, the design parameters obtained from experiments can be expressed as intervals under a certain confidence probability [14]. The interval form can effectively quantify the uncertainty of design parameters, and the measured truth value is usually included in this interval. The interval number is used to express the value range of design parameters, and the interval analysis theory is used to calculate the design index. This method can effectively reflect the essence of the complexity of design engineering. Therefore, the proposed interval calculation method suitable for road design theory will be beneficial for the reliability analysis of new pavement structure design.

Interval analysis is developed from the error theory of computational mathematics. It is a mathematical branch that uses interval variables instead of point variables [15]. In engineering application, interval analysis theory can not only solve the problem of insufficient precision of traditional point numerical calculation models, but also optimize the calculation result and provide an alternative method for the accurate application of the engineering calculation model [16,17,18]. Tang et al. [19] expressed the fatigue life of cement stabilized stone in intervals, and obtained the fatigue interval equation of 95% guarantee rate, which greatly improved the efficiency of the test data. Impollonia et al. [20] proposed an interval assessment method for the static response of axial stiffness structures, which overcomes the defects caused by dependence in traditional analysis. Xie et al. [21] used the interval parameters in order to establish the fatigue characteristics model under different stress states, and realized the fatigue characteristic standardization model based on the new interval analysis method, which solved the problems of insufficient sampling performance, low precision and insufficient stability of test equipment. In order to realize the calculation of parameters with boundaries but uncertain sizes, Guerine et al. [22] proposed an interval analysis method of dynamic response from a wind turbine gear system; Long et al. [23] proposed an interval analysis method of fatigue crack growth life prediction; and Liu et al. [24] proposed an interval uncertainty analysis method of structural static response. Galván et al. [25] obtained the best solution when designing dynamic systems by introducing interval analysis theory. Viegas et al. [26] used interval analysis to solve the two limitations of the parallel motion machine design method. However, while interval analysis theory provides new ideas in engineering applications, due to the laziness of interval variables to upper and lower limits in the calculation process, there are cases where the interval results of theoretical calculations are inconsistent with actual engineering applications. 

In order to solve the problem that interval analysis is not applicable due to interval expansion, when the finite element method is used to solve the structural mechanics interval, the calculation theory of interval transformation has been proposed by many researchers [27,28,29,30,31]. However, the application of interval analysis theory has been hindered in the road design process. On the one hand, the calculation of interval models is very dependent on computer software, which makes interval analysis theory difficult to promote in road engineering; on the other hand, the results of interval expansion cannot be matched with the actual situation of the project, resulting in the calculation results deviating from the scope of engineering application. In view of this, this paper takes the calculation model of the permanent deformation of asphalt mixture as the analysis object and deduces the interval calculation formula. Combining the calculation principle of interval theory and considering the engineering application range of design parameters, an interval calculation method based on point arithmetic is proposed. In addition, in order to avoid the occurrence of interval expansion, an analysis method based on interval variable weight allocation is proposed.

## 2. Interval Analysis Theory

In 1962, the theory of interval analysis was proposed by the American mathematician Moore [32]. The theory is to improve the conventional point numerical operation into interval operation, which can realize the calculation of bounded but uncertain size data. Since the theory of interval analysis was proposed, many scholars have studied this branch of mathematics. The most significant work is Moore’s monograph “Interval Analysis” published in 1966 [33], which laid a foundation for the development of interval analysis theory.

### 2.1. Basic Representation of the Interval

A contiguous subset $X=\left[\underline{X},\bar{X}\right]$ on the real set R is called a real interval.

When $\underline{X}=\bar{X}$, it is called the degradation interval and represents a certain real number.

The set of all real intervals is denoted as $IR=\{\left[\underline{X},\bar{X}\right]:\underline{X},\bar{X}\in R,\underline{X}\leq \bar{X}\}$.

The upper and lower endpoints of the interval X are denoted as sup(X) and inf(X), respectively.

The midpoint, width, radius and absolute value of interval X are defined as follows:(1)$$\mathrm{Midpoint}:\mathrm{mid}\left(X\right)=\left(\underline{X}+\bar{X}\right)/2$$
(2)$$\mathrm{Width}:\mathrm{wid}\left(X\right)=\bar{X}-\underline{X}$$
(3)$$\mathrm{Radius}:\mathrm{rad}\left(X\right)=\left(\bar{X}-\underline{X}\right)/2$$
(4)$$\mathrm{Absolute}\;\mathrm{Value}:\left|X\right|=max\{\left|\underline{X}\right|,\left|\bar{X}\right|\}$$

A short note: m(X)=mid(X),w(X)=wid(X),r(X)=rid(X).

### 2.2. The Basic Algorithm of the Interval

Let $X=\left[\underline{X},\bar{X}\right],Y=\left[\underline{Y},\bar{Y}\right]\in IR$. The four arithmetic rules of the interval are: (5)$$X+Y=\left[\underline{X}+\underline{Y},\bar{X}+\bar{Y}\right]$$
(6)$$X-Y=\left[\underline{X}-\bar{Y},\bar{X}-\underline{Y}\right]$$
(7)$$X\times Y=\left[{\left(\underline{X}\underline{Y},\underline{X}\bar{Y},\bar{X}\underline{Y},\bar{X}\bar{Y}\right)}_{min},{\left(\underline{X}\underline{Y},\underline{X}\bar{Y},\bar{X}\underline{Y},\bar{X}\bar{Y}\right)}_{max}\right]$$
(8)$$X÷Y=\left[\underline{X},\bar{X}\right]\times \left[\frac{1}{\bar{Y}},\frac{1}{\underline{Y}}\right],0\notin Y$$

## 3. Interval Calculation of Permanent Deformation of Asphalt Mixture

### 3.1. Calculation Formula of Permanent Deformation Interval

According to the calculation method of asphalt pavement design index [1], the interval calculation formula applicable to the permanent deformation of asphalt mixture is derived.

Interval Calculation Formula for the Average Daily Equivalent Axis of the Design Year of the Initial Year

When calculating the average daily number of large passenger cars and trucks, AADT statistics have a large uncertainty [6]. The AADT expressed as an interval is more reasonable. Therefore, the interval calculation formula for the average daily equivalent axis of the design year of the initial year can be expressed as:(9)$$\left[\underline{N_{1}},\bar{N_{1}}\right]=\left[\underline{AADTT},\bar{AADTT}\right]\times DDF\times LDF\times \overset{11}{\underset{m=2}{\sum }}\left(VCDF_{m}\times EALF_{m}\right)$$
where $\left[\underline{N_{1}},\bar{N_{1}}\right]$ is the daily average equivalent axis subsection of the designed lane in the initial year (times); $\left[\underline{AADTT},\bar{AADTT}\right]$ is the bidirectional annual average daily traffic interval for 2-axle 6-wheel and above vehicles (vehicles/d); DDF is the direction coefficient; LDF is the lane coefficient; m is the vehicle type number; $VCDF_{m}$ is a class m vehicle type assignment coefficient; and $EALF_{m}$ is the equivalent design axle load conversion factor for class m vehicles.

2.Interval Calculation Formula for the Cumulative Number of Times of the Equivalent Design Axle Load

The average annual growth rate of traffic volume within the design life is obviously a value that cannot be estimated with accuracy, and it is more consistent with the actual situation to express it in the form of a certain range of change. The interval calculation formula for the cumulative number of times of the equivalent design axle load is:(10)$$\left[\underline{N_{e}},\bar{N_{e}}\right]=\frac{365\times \left[\underline{N_{1}},\bar{N_{1}}\right]\{{\left(1+\left[\underline{\gamma },\bar{\gamma }\right]\right)}^{t}-1\}}{\left[\underline{\gamma },\bar{\gamma }\right]}$$
where $\left[\underline{N_{e}},\bar{N_{e}}\right]$ is the equivalent design axle load cumulative action frequency interval (times); *t* is the design period (year); and $\left[\underline{\gamma },\bar{\gamma }\right]$ is the average annual growth interval of traffic volume within the design period (%).

3.Interval Calculation Formula for Permanent Deformation

(11)$$\left[\underline{R_{ai}},\bar{R_{ai}}\right]=2.31\times 10^{-8}k_{Ri}T_{pef}^{2.93}p_{i}^{1.80}{\left[\underline{N_{e}},\bar{N_{e}}\right]}^{0.48}\left(\frac{h_{i}}{h_{0}}\right)\left[\underline{R_{0i}},\bar{R_{0i}}\right]$$
where $\left[\underline{R_{ai}},\bar{R_{ai}}\right]$ is the interval of the *i*-th layer permanent deformation of the asphalt mixture (mm); $k_{Ri}$ is the comprehensive correction coefficient; $p_{i}$ is the vertical compressive stress of the *i*-th layer top surface of the asphalt mixture layer (MPa); $T_{pef}$ is the equivalent temperature of the permanent deformation of the asphalt mixture layer (°C); $h_{i}$ is the *i*-th layer thickness (mm); $h_{0}$ is the thickness of the rutting test piece (mm); $\left[\underline{R_{0i}},\bar{R_{0i}}\right]$ is the *i*-th layer asphalt mixture at a test temperature of 6 °C, a pressure of 0.7 MPa, and the number of loadings is 2520 times, the rutting test permanent deformation interval (mm).

### 3.2. Determination of Calculation Parameters for Permanent Deformation

The highway with a design period of 15 years has an asphalt surface type of AC-16. According to the analysis of the traffic survey, the average growth rate of the traffic volume within the designed service life is around 5~6%, the DDF is 0.55, the LDF is 0.7, and the AADTT is 7064 vehicles/d. According to reference [6], the relative uncertainty of AADTT can be taken as 6.5%, then $\left[\underline{AADTT},\bar{AADTT}\right]$ = [6605, 7523] (vehicles/d). According to Equation (12), the accumulated traffic volume of large passenger cars and trucks within the designed service life of the highway is [20.8, 23.7] (106 vehicles). As is evident from Table 1, the traffic grade of the highway designs belongs to “Especially Heavy” traffic. According to the traffic historical data, it is determined that the highway is a TTC3 class, and the sum of the calculated product of $VCDF_{m}$ and $EALF_{m}$ is:$\overset{11}{\underset{m=2}{\sum }}\left(VCDF_{m}\times EALF_{m}\right)=6.76$.
(12)$$N^{\prime }=AADTT\times DDF\times LDF\times \frac{365\times \{{\left(1+\gamma \right)}^{t}-1\}}{\gamma }$$
where $N^{\prime }$ is the total number of passenger cars and trucks designed to design lanes within the design life (106 vehicles).

The asphalt mixture pavement structure is layered according to the calculation requirements of the permanent deformation. This article takes the first layer of asphalt mixture as an example, and the layer thickness is $h_{1}=z_{1}=15\mathrm{mm}$. The thickness of the rutting test piece is $h_{a}=50\mathrm{mm}$. The other layers can refer to the calculation process of the first layer. In determining the comprehensive correction coefficient:$$d_{1}=-1.35\times 10^{-4}h_{a}^{2}+8.18\times 10^{-2}h_{a}-14.5=-14.4853\;d_{2}=8.78\times 10^{-7}h_{a}^{2}-1.5\times 10^{-3}h_{a}+0.9=0.8985\;k_{R1}=\left(d_{1}+d_{2}+z_{1}\right)\times 0.9731^{z_{1}}=0.9475$$

$p_{1}$ is 0.7 MPa and $T_{pef}$ is 23.8 °C. When the asphalt mixture was loaded 2520 times, the permanent deformation of the measured rut test of the AC-16 mixture was 0.628 mm, see Figure 1 for the rut test. Considering the uncertainty introduced by the measurement deformation, the measurement uncertainty of the deformation is calculated according to the theory of measurement uncertainty. The results are shown in Table 2. Then, the deformation range of the AC-16 mixture when loaded 2520 times is $\left[\underline{R_{0i}},\bar{R_{0i}}\right]$ = [0.626, 0.630] (mm).

### 3.3. Calculation of Permanent Deformation Interval

According to Equation (9), using the INTLAB developed by Professor Rump of the Technical University of Hamburg, $\left[\underline{N_{1}},\bar{N_{1}}\right]=$ [17,191, 19,580] (times/d). The calculation code is as follows:


*>>AADTT = infsup(6605, 7523);DDF = 0.55;LDF = 0.7;VE = 6.76;*



*int_N1 = AADTT × DF × LDF × VE*



*intval int_N1 = 1.0 × 10^4^ × [1.7191, 1.9580]*


According to Equation (10), $\left[\underline{N_{e}},\bar{N_{e}}\right]=\left[1.1282,1.9961\right]$(10^8^ times). The calculation code is as follows:


*>>r = infsup(0.05,0.06); t = 15; int_Ne = 365 × int_N1 × ((1 + r)^t−1)/r*



*intval int_Ne = 1.0 × 10^8^ × [1.1282, 1.9961]*


It is easy to find that there exists a phenomenon of interval expansion of $\left[\underline{N_{e}},\bar{N_{e}}\right]$. According to Equation (12), the cumulative traffic volume of large passenger cars and trucks in the design lanes calculated by the expansion results is [16.7, 29.5] (106 vehicles). In reference to Table 1, the inspection result interval exceeds the engineering requirements of the highway “Especially Heavy” traffic design level. Therefore, the result of $\left[\underline{N_{e}},\bar{N_{e}}\right]$ cannot be used as the design parameter of the highway.

In order to solve the problem that the calculation results did not accord with the engineering application caused by interval expansion, the calculation relation of interval expansion is taken as the analysis object, and the interval calculation relation is simplified as the idea of replacing interval calculation with interval endpoint value. The adjustment method is as follows: When the formula has subtraction or division operations between intervals, and it is verified that the interval results obtained by this step do not meet the requirements of engineering applications, then, the subtraction or division between the intervals should be simplified to the subtraction or division of the upper and lower end points of the two intervals, and the calculation results should be allocated according to the size, which are used as the upper and lower end points of the calculation amount interval results, respectively.

According to the adjustment method of eliminating the interval expansion, the interval result calculated by INTLAB is $\left[\underline{N_{e}},\bar{N_{e}}\right]=$ [1.3539, 1.6635] (10^8^ times). The calculation code is as follows:


*>>N11 = inf(int_N1);N12 = sup(int_N1);r1 = inf(r);r2 = sup(r);Ne1 = 365×N11×((1 + r1)^t−1)/r1;Ne2 =365×N12×((1 + r2)^t−1)/r2;int_Ne = infsup(Ne1,Ne2)*



*intval int_Ne = 1.0 × 10^8^ × [1.3539, 1.6635]*


According to Equation (12), [$\underline{N^{\prime }},\bar{N\prime }$] = [20.0, 24.6] (106 vehicles). Referring to Table 1, the inspection result interval is in line with the engineering requirements of the highway “Especially Heavy” traffic design level. Therefore, after adjusting to eliminate the interval extension, $\left[\underline{N_{e}},\bar{N_{e}}\right]$ can be applied to the design of this highway.

According to Equation (11), $\left[\underline{N_{e}},\bar{N_{e}}\right]=$ [1.3539, 1.6635] (10^8^ times) is taken as the interval variable, and the interval result calculated by INTLAB turns out to be $\left[\underline{R_{ai}},\bar{R_{ai}}\right]$ = [0.1868, 0.2077] (mm). The calculation code is as follows:


*>>Kr1 = 0.9475;p1 = 0.7;Tpef = 23.8;h1 = 15;h0 = 50;R01 = infsup(0.626,0.630);int_Ra1 = 2.31×10^−8×Kr1×Tpef^2.93×p1^1.8×int_Ne^0.48×(h1/h0)× R01*



*intval int_Ra1 = [0.1868, 0.2077]*


## 4. Calculation of Permanent Deformation of Asphalt Mixture Based on Point Algorithm

### 4.1. Interval Simplification Calculation Method Based on Point Algorithm

Based on the research content of this article, the mathematical conditions for realizing point value instead of interval calculation are as follows: The range of interval variables is positive, and the mathematical operation relation is the addition, subtraction, multiplication, division between interval variables and point values or interval variables, and the power of interval variables and constants. Analyzing the four arithmetic rules of intervals, it is easy to find that the addition of intervals is easy to subtract, and the division of intervals is essentially a conversion form of interval multiplication. Therefore, in order to simplify the analysis process of the point numerical calculation interval formula, it is proposed to use only two operations of addition and multiplication to realize the simplified calculation of the interval. Based on the above analysis ideas, the simplified principle of the calculation interval formula based on the point operation rule is:When there is no mutual subtraction or division of interval variables in the interval calculation formula:(1)If there is no interval variable in the interval calculation formula as the denominator or the interval variable contains a negative value power, then the lower end point values of all interval variables are substituted into the numerical calculation formula of the point to calculate an end point value of the interval result. At the same time, the upper end point values of all interval variables are substituted into the point numerical calculation formula, and the other end point value of the interval result is calculated. Comparing the size of the two end points, the expression in the form of an interval is the calculation result of the calculation formula of the interval. Such as:$$\left[\underline{Y},\bar{Y}\right]=a\times \left[\underline{X_{1}},\bar{X_{1}}\right]+\left[\underline{X_{2}},\bar{X_{2}}\right]\;\underline{Y}=a\times \underline{X_{1}}+\underline{X_{2}};\bar{Y}=a\times \bar{X_{1}}+\bar{X_{2}}$$(2)If the interval calculation formula contains the interval variable as the denominator or the interval variable contains the negative point value power, then the following form transformation needs to be made to the interval variable:Let $X=\left[\underline{X},\bar{X}\right]\in IR,a\in R\;\mathrm{and}\;a>0$. When the interval variable X is used as the denominator of the constant a, its form for point numerical calculation is transformed into:(13)$$\frac{a}{\left[\underline{X},\bar{X}\right]}=a\times \left[\frac{1}{\bar{X}},\frac{1}{\underline{X}}\right]$$
when the interval variable X contains a negative power of degree a, its form for point value calculation is transformed into:(14)$${\left[\underline{X},\bar{X}\right]}^{-a}={\left[\frac{1}{\bar{X}},\frac{1}{\underline{X}}\right]}^{a}$$After the interval variable is transformed, it can be calculated according to step (1).When there is mutual subtraction or division of interval variables in the interval calculation formula, it should be judged whether the expansion result of the interval meets the engineering requirements.(1)When the interval expansion result meets the engineering requirements, the following form transformation can be made in the section where the interval expansion exists in the interval calculation formula:Let $X=\left[\underline{X},\bar{X}\right],Y=\left[\underline{Y},\bar{Y}\right]\in IR$: When the interval extension calculation formula is X−Y, the form transformation method is:(15)$$X-Y=\left[\underline{X},\bar{X}\right]-\left[\underline{Y},\bar{Y}\right]=\left[\underline{X},\bar{X}\right]+\left[-\bar{Y},-\underline{Y}\right]=\left[\underline{X}-\bar{Y},\bar{X}-\underline{Y}\right]$$
when the interval extension calculation formula is X/Y, the form transformation method is:(16)$$\frac{X}{Y}=\frac{\left[\underline{X},\bar{X}\right]}{\left[\underline{Y},\bar{Y}\right]}=\left[\underline{X},\bar{X}\right]\times \left[\frac{1}{\bar{Y}},\frac{1}{\underline{Y}}\right]=\left[\frac{\underline{X}}{\bar{Y}},\frac{\bar{X}}{\underline{Y}}\right]$$After adjusting the interval variable according to the above method, it can be calculated according to the method of the first simplified principle.(2)When the interval expansion result does not meet the engineering requirements, the interval calculation rule needs to be adjusted. Taking the calculation of the interval expansion as the analysis object, the interval calculation is then simplified to the idea of replacing the interval with the endpoint value. The adjustment methods of the subtraction and division between the intervals are as follows:
(17)$$X-Y=\left[\underline{X},\bar{X}\right]-\left[\underline{Y},\bar{Y}\right]=\left[\underline{X}-\underline{Y},\bar{X}-\bar{Y}\right]$$
(18)$$\frac{X}{Y}=\frac{\left[\underline{X},\bar{X}\right]}{\left[\underline{Y},\bar{Y}\right]}=\left[\frac{\underline{X}}{\underline{Y}},\frac{\bar{X}}{\bar{Y}}\right]$$After the interval variable is adjusted according to the above method, it can be calculated according to the method of the first simplified principle.

### 4.2. Interval Calculation of Permanent Deformation of Asphalt Mixture Based on Point Algorithm

There is no interval expansion or interval form conversion in Equation (9), so according to point (1) of Article 1 in the simplified principle:$$\underline{N_{1}}=\underline{AADTT}\times DDF\times LDF\times \overset{11}{\underset{m=2}{\sum }}\left(VCDF_{m}\times EALF_{m}\right)=17,190\left(\mathrm{times}/d\right)\;\bar{N_{1}}=\bar{AADTT}\times DDF\times LDF\times \overset{11}{\underset{m=2}{\sum }}\left(VCDF_{m}\times EALF_{m}\right)=19,580\left(\mathrm{times}/d\right)$$

Therefore, the interval result calculated according to the point value is [17,190, 19,580] (times/d).

There is a case of interval expansion in Equation (10), and the interval result obtained does not meet the requirements of engineering applications. According to point (2) of Article 2 of the simplification principle, it is necessary to adjust the calculation rule of the interval expansion variable first, and then simplify the calculation of the point value. 

Interval variable form transformation formula:(19)$$\left[\underline{N_{e}},\bar{N_{e}}\right]=365\left[\underline{N_{1}},\bar{N_{1}}\right]\{{\left(1+\left[\underline{\gamma },\bar{\gamma }\right]\right)}^{t}-1\}\times \left[\frac{1}{\bar{\gamma }},\frac{1}{\underline{\gamma }}\right]$$
$$\underline{N_{e}}=365\times \underline{N_{1}}\{{\left(1+\underline{\gamma }\right)}^{t}-1\}\times \frac{1}{\underline{\gamma }}=1.3539\left(10^{8}\mathrm{times}\right)$$
$$\bar{N_{e}}=365\times \bar{N_{1}}\{{\left(1+\bar{\gamma }\right)}^{t}-1\}\times \frac{1}{\bar{\gamma }}=1.6634\left(10^{8}\mathrm{times}\right)$$

Therefore, the interval result calculated from the point value is [1.3539, 1.6634] (10^8^ times).

There is neither interval expansion nor interval form conversion requirements in Equation (11), so it can be calculated according to point (1) of Article 1 in the simplified principle.
$$\underline{R_{a1}}=2.31\times 10^{-8}\times k_{Ri}\times T_{pef}^{2.93}\times p_{i}^{1.80}\times {\underline{N}}_{e}^{0.48}\times \left(\frac{h_{i}}{h_{0}}\right)\times \underline{R_{0i}}=0.1869\left(\mathrm{mm}\right)\;\bar{R_{a1}}=2.31\times 10^{-8}\times k_{Ri}\times T_{pef}^{2.93}\times p_{i}^{1.80}\times {\bar{N}}_{e}^{0.48}\times \left(\frac{h_{i}}{h_{0}}\right)\times \bar{R_{0i}}=0.2076\left(\mathrm{mm}\right)$$

Therefore, the interval result calculated from the point value is [0.1869, 0.2076] (mm).

The results of applying INTLAB and the interval calculation based on point values are summarized in Table 3.

It can be seen from Table 3 that, ignoring the transfer error of the cumulative calculation of the effective number of digits of the calculation tool, it can be concluded that the result interval of the point value calculation is completely consistent with the result of the INTLAB code calculation. The feasibility and effectiveness of simplifying the calculation interval formula based on point values are proved.

## 5. Interval Simplification Calculation Based on Interval Variable Weight Assignment

### 5.1. Simplified Analysis Method Based on Interval Variable Weight Assignment

In order to avoid the interval expansion of multi-interval variable formulas, this section proposes a simplified method based on interval variable weight assignment, which takes multiple interval results as the analysis object. First, the calculation formula with multiple interval variables at the same time is split into multiple interval calculation formulas with only one interval variable. Second, a plurality of different interval variables is reasonably weighted. Lastly, the weight is multiplied by the corresponding interval variable calculation result, and the total interval result is obtained, that is, the interval result of the calculation amount.

The above simplified method is represented mathematically, and the analysis steps are as follows:Split Multi-interval Variable Formula

Let $X=\left[\underline{X},\bar{X}\right],X_{1}=\left[\underline{X_{1}},\bar{X_{1}}\right],X_{2}=\left[\underline{X_{2}},\bar{X_{2}}\right],X_{3}=\left[\underline{X_{3}},\bar{X_{3}}\right]\in IR$ and $x_{1}=mid\left(X_{1}\right),x_{2}=mid\left(X_{2}\right),x_{3}=mid\left(X_{3}\right)$. Existing interval calculation formula:(20)$$\left[\underline{X},\bar{X}\right]=\frac{\left[\underline{X_{1}},\bar{X_{1}}\right]-\left[\underline{X_{2}},\bar{X_{2}}\right]}{\left[\underline{X_{3}},\bar{X_{3}}\right]}$$

The corresponding interval variable is replaced by the interval midpoint, and the multi-interval variable formula is divided into three interval calculation formulas with only one interval variable:(21)$${\left[\underline{X},\bar{X}\right]}_{1}=\frac{\left[\underline{X_{1}},\bar{X_{1}}\right]-x_{2}}{x_{3}}$$
(22)$${\left[\underline{X},\bar{X}\right]}_{2}=\frac{x_{1}-\left[\underline{X_{2}},\bar{X_{2}}\right]}{x_{3}}$$
(23)$${\left[\underline{X},\bar{X}\right]}_{3}=\frac{x_{1}-x_{2}}{\left[\underline{X_{3}},\bar{X_{3}}\right]}$$

2.Calculate the Weight of Each Interval Variable

The weight of the interval variable may be different in different engineering projects, and the calculation of the weight should be comprehensively determined in accordance with the actual engineering conditions. The combination weighting method has the advantages of subjective and objective weighting methods. According to reference [34], the calculation formula of weights is:(24)$$q_{i}=aw_{i}^{\prime }+bw_{i}^{\prime \prime }$$
where $q_{i}$ is the integrated weight of the i-th interval variable; $w_{i}^{\prime }$ is the subjective weight of the *i*-th interval variable; $w_{i}^{\prime \prime }$ is the objective weight of the *i*-th interval variable; *a*, *b* is the evaluator’s degree of trust in subjective and objective weights, *a* + *b* = 1.

3.Calculate the Interval Result of Each Single Interval Variable Formula

The single interval variable formula can easily use the point algorithm to calculate the interval result. Taking the single interval variable formula split by Equation (20) as an example, the point value calculation formulas of each single interval variable are:(1)There is no single-interval variable form transformation in Equation (21), and the interval result endpoint value can be directly calculated as follows:
$$D_{1}=\frac{\underline{X_{1}}-x_{2}}{x_{3}};D_{2}=\frac{\bar{X_{1}}-x_{2}}{x_{3}}$$Compare the size of *D*_1_ and *D*_2_ to determine the interval result of Equation (21).(2)The interval variable in Equation (22) is used as a subtraction. After the interval variable form is transformed, the calculation can be simplified directly according to the point algorithm. The calculation method is as follows:
(25)$${\left[\underline{X},\bar{X}\right]}_{2}=\frac{x_{1}-\left[\underline{X_{2}},\bar{X_{2}}\right]}{x_{3}}=\frac{x_{1}+\left[-\bar{X_{2}},-\underline{X_{2}}\right]}{x_{3}}$$Therefore, $D_{1}=\frac{x_{1}+\left(-\bar{X_{2}}\right)}{x_{3}};D_{2}=\frac{x_{1}+\left(-\underline{X_{2}}\right)}{x_{3}}$Compare the size of *D*_1_ and *D*_2_ to determine the interval result of Equation (22).(3)The interval variable in Equation (23) is the divisor. After the interval variable form is transformed, the calculation can be simplified directly according to the point algorithm. The calculation method is as follows: (26)$${\left[\underline{X},\bar{X}\right]}_{3}=\frac{x_{1}-x_{2}}{\left[\underline{X_{3}},\bar{X_{3}}\right]}=\left(x_{1}-x_{2}\right)\times \left[\frac{1}{\bar{X_{3}}},\frac{1}{\underline{X_{3}}}\right]$$Therefore, $D_{1}=\left(x_{1}-x_{2}\right)\times \frac{1}{\bar{X_{3}}};D_{2}=\left(x_{1}-x_{2}\right)\times \frac{1}{\underline{X_{3}}}$Compare the size of *D*_1_ and *D*_2_ to determine the interval result of Equation (23).

4.The Result of the Interval of the Synthetic Calculation

If the interval variables $X_{1},X_{2}$ and $X_{3}$ determined in step 2 have weights of $q_{1},q_{2}$ and $q_{3}$ and $q_{1}+q_{2}+q_{3}=1$, respectively, and the result of the calculation formula of the single interval variable calculated in step 3 based on the point algorithm is ${\left[\underline{X},\bar{X}\right]}_{1},{\left[\underline{X},\bar{X}\right]}_{2}$ and ${\left[\underline{X},\bar{X}\right]}_{3}$, then the interval result of the calculation amount X is synthesized as:(27)$$\left[\underline{X},\bar{X}\right]=q_{1}{\left[\underline{X},\bar{X}\right]}_{1}+q_{2}{\left[\underline{X},\bar{X}\right]}_{2}+q_{3}{\left[\underline{X},\bar{X}\right]}_{3}$$

The method of interval variable weight assignment can effectively avoid the complicated interval expansion adjustment process, and the split calculation formula containing only one interval variable is simpler to calculate with the help of the point value. The weighted assignment of interval variables is used to synthesize the interval results of the calculation amount, so that the importance of each interval variable under different engineering backgrounds can be flexibly adjusted, thereby calculating the interval results that are most in line with the actual project.

### 5.2. Case Analysis Based on Interval Variable Weight Assignment Method

Taking Equation (10) as a case, split Equation (10) into the following three single-interval variable formulas:(28)$${\left[\underline{N_{e}},\bar{N_{e}}\right]}_{1}=\frac{365\left[\underline{N_{1}},\bar{N_{1}}\right]\{{\left(1+\gamma \right)}^{t}-1\}}{\gamma }$$
(29)$${\left[\underline{N_{e}},\bar{N_{e}}\right]}_{2}=\frac{365N_{1}\{{\left(1+\left[\underline{\gamma },\bar{\gamma }\right]\right)}^{t}-1\}}{\gamma }$$
(30)$${\left[\underline{N_{e}},\bar{N_{e}}\right]}_{3}=\frac{365N_{1}\{{\left(1+\gamma \right)}^{t}-1\}}{\left[\underline{\gamma },\bar{\gamma }\right]}=365N_{1}\{{\left(1+\gamma \right)}^{t}-1\}\times \left[\frac{1}{\bar{\gamma }},\frac{1}{\underline{\gamma }}\right]$$

The above three single-interval variable calculation formulas do not need to consider the case of interval expansion, and only the corresponding form transformation can be performed on the interval variable divisor in Equation (30). According to Equation (24), in order to prove the flexibility of the weight assignment of interval variables of the same calculation model in different projects, this case provides two calculation schemes of weight assignment. The synthetic weight of variables $N_{1}$ and γ is calculated according to the evaluation parameters shown in Table 4:

According to the calculation parameters determined in this paper, combined with Equations (28)–(30), the interval results can be easily calculated based on the point algorithm, and the simplified calculation process is as follows:(1)Calculation of the endpoint value of Equation (28)
$${\underline{N_{e}}}_{1}=\frac{365\times \underline{N_{1}}\times \{{\left(1+\gamma \right)}^{t}-1\}}{\gamma }=1.4060\left(10^{8}\mathrm{times}\right)\;{\bar{N_{e}}}_{1}=\frac{365\times \bar{N_{1}}\times \{{\left(1+\gamma \right)}^{t}-1\}}{\gamma }=1.6015\left(10^{8}\mathrm{times}\right)$$

Therefore, ${\left[\underline{N_{e}},\bar{N_{e}}\right]}_{1}=\left[1.4060,1.6015\right]\left(10^{8}\mathrm{times}\right)$.

(2)Calculation of the endpoint value of Equation (29)

$${\underline{N_{e}}}_{2}=\frac{365\times N_{1}\times \{{\left(1+\underline{\gamma }\right)}^{t}-1\}}{\gamma }=1.3164\left(10^{8}\mathrm{times}\right)\;{\bar{N_{e}}}_{2}=\frac{365\times N_{1}\times \{{\left(1+\bar{\gamma }\right)}^{t}-1\}}{\gamma }=1.7039\left(10^{8}\mathrm{times}\right)$$

Therefore, ${\left[\underline{N_{e}},\bar{N_{e}}\right]}_{2}=\left[1.3164,1.7039\right]\left(10^{8}\mathrm{times}\right)$.

(3)Calculation of the endpoint value of Equation (30)

$${\underline{N_{e}}}_{3}=365\times N_{1}\times \{{\left(1+\gamma \right)}^{t}-1\}\times \frac{1}{\bar{\gamma }}=1.3784\left(10^{8}\mathrm{times}\right)\;{\bar{N_{e}}}_{3}=365\times N_{1}\times \{{\left(1+\gamma \right)}^{t}-1\}\times \frac{1}{\underline{\gamma }}=1.6541\left(10^{8}\mathrm{times}\right)$$

Therefore, ${\left[\underline{N_{e}},\bar{N_{e}}\right]}_{3}=\left[1.3784,1.6541\right]\left(10^{8}\mathrm{times}\right)$.

According to Table 4, scheme 1 is that the weight of variable $N_{1}$ accounts for 60%, and the weight of γ accounts for 40%. Scheme 2 is that the weight of the variable $N_{1}$ accounts for 40%, and the weight of γ accounts for 60%. In both schemes, the weight of the growth rate γ as the numerator and denominator accounts for 50% of the allocated weight. According to Equation (27), the results of the two schemes are:

Scheme 1:$$\left[\underline{N_{e}},\bar{N_{e}}\right]=0.6\times {\left[\underline{N_{e}},\bar{N_{e}}\right]}_{1}+0.2\times {\left[\underline{N_{e}},\bar{N_{e}}\right]}_{2}+0.2\times {\left[\underline{N_{e}},\bar{N_{e}}\right]}_{3}=\left[1.3826,1.6325\right]\left(10^{8}\mathrm{times}\right)$$

Scheme 2:$$\left[\underline{N_{e}},\bar{N_{e}}\right]=0.4\times {\left[\underline{N_{e}},\bar{N_{e}}\right]}_{1}+0.3\times {\left[\underline{N_{e}},\bar{N_{e}}\right]}_{2}+0.3\times {\left[\underline{N_{e}},\bar{N_{e}}\right]}_{3}=\left[1.3708,1.6480\right]\left(10^{8}\mathrm{times}\right)$$

The interval results calculated according to the weight assignment of scheme 1 and scheme 2 are [1.3826, 1.6325] (10^8^ times) and [1.3708, 1.6480] (10^8^ times), respectively. It is easy to know that the results of $\left[\underline{N_{e}},\bar{N_{e}}\right]$ with $N_{1}$ or γ as the main weights are similar, indicating that the equivalent design cumulative axis times are quite sensitive to the interval variables $N_{1}$ and γ. The different method to calculate $\left[\underline{N_{e}},\bar{N_{e}}\right]$ summary is shown in Table 5.

It can be seen from Table 5 that the results of the three intervals are similar. According to Equation (12), scheme 1: [$\underline{N^{\prime }},\bar{N^{\prime }}$] = [20.5, 24.1] (106 vehicles); scheme 2: [$\underline{N^{\prime }},\bar{N^{\prime }}$] = [20.3, 24.4] (106 vehicles). In reference to Table 1, the interval results calculated by the two schemes are in line with the engineering background of the design level of the “Especially Heavy” traffic load. According to Equation (11), based on the simplified rule of point operation, the calculation of the permanent deformation interval of the asphalt mixture under the two schemes can be achieved. The permanent deformations calculated by the two schemes based on the point operation rule and the interval variable weight assignment method are summarized. The results are shown in Table 6.

According to the analysis in Table 6, it can be seen that under three different calculation methods, the permanent deformation interval results of asphalt mixture are all similar. Although the permanent deformation interval calculated based on the interval variable weight assignment is not completely consistent with the result calculated by the INTLAB code, it meets the application requirements in engineering. 

To summarize, this paper proposes two simplified methods for calculating the interval of permanent deformation of asphalt mixtures—a calculation method based on point arithmetic and a calculation method based on interval variable weight assignment. Among them, on the basis of the adjustment calculation process, the multi-interval variable calculation formula can be solved at one time, and the calculation speed is faster, but the adjustment process has certain complexities. The method based on the weight assignment of interval variables can effectively avoid the interval expansion of the calculation formula of multi-interval variables, reduce the complex operation of multi-interval variables to the calculation of single interval variables, and greatly simplify the analysis process of interval calculation, but the workload of interval calculation has increased. The method based on the weight assignment of interval variables is more suitable for the use of new scholars and the majority of grassroots engineering operators. In addition, this method can also flexibly adjust the weight of each interval variable according to the situation of different projects, and can be used as a tool to analyze variables that have a significant impact on the project calculation index. Briefly, the above two simplified methods that apply the interval analysis theory to the engineering field are feasible and reliable, and can achieve the goal of simplifying the calculation of the interval engineering theory. When using interval analysis to solve actual engineering problems, it can provide a reference for engineering personnel.

## 6. Conclusions

In this paper, the permanent deformation of asphalt mixture is taken as an example, and combined with the interval analysis theory, the simplified analysis method of the calculation formula of the permanent deformation is mainly studied. This study draws the following conclusions.

(1)The study found that in the calculation formula of multi-interval variables, the subtraction or division of the two intervals will expand the interval calculation results. In order to ensure the engineering validity of the calculation results, the calculation results should be verified.(2)Combining the four arithmetic rules of intervals, the two-point numerical operation of addition and multiplication can realize the simplified calculation of an interval formula. This method avoids the complicated software application process and is suitable for the grassroots staff.(3)The simplified calculation method based on the weight assignment of interval variables can effectively avoid the occurrence of interval expansion. Taking Equation (10) in this article as an analysis case, the effectiveness of this method is revealed. This method has more room to play when calculating multi-interval variable formulas.

## Figures and Tables

**Figure 1 materials-14-02116-f001:**
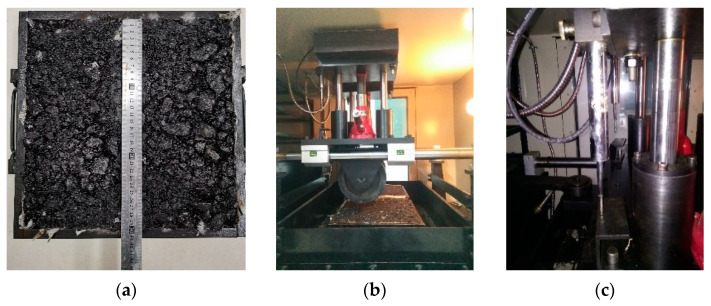
(**a**) Rutting test piece, (**b**) rutting test, (**c**) displacement sensor.

**Table 1 materials-14-02116-t001:** Design traffic load level.

Design Traffic Load Level	Extremely Heavy	Especially Heavy	Weight	Medium	**Light**
$N^{\prime }$(10^6^vehicles)	≥50.0	50.0~19.0	19.0~8.0	8.0~4.0	<4.0

**Table 2 materials-14-02116-t002:** Summary of uncertainty components of deformation.

Input Quantity, $X_{i}$	Source of Uncertainty	$\mathrm{Standard}\;(\mathrm{Relative})\;\mathrm{Uncertainty},\;u\left(x_{i}\right)$	Expected Value	$\mathrm{Relative}\;\mathrm{Uncertainty}\;\mathrm{Component},\;u_{rel}\left(x_{i}\right)$	$\mathrm{Synthetic}\;\mathrm{Relative}\;\mathrm{Uncertainty},\;u_{crel}\left(y_{i}\right)$
Deformation	Indication error	0.289%	—	0.289%	0.294%
	Minimum resolution	0.00289 mm	1.154 mm	0.025%	
	Sensor error	0.05%	—	0.05%	

**Table 3 materials-14-02116-t003:** Summary of interval results for different calculation methods.

**Interval Calculation** **Index**	INTLAB Interval Calculation Results	Interval Results Based on Point Numerical Calculation
$\left[\underline{N_{1}},\bar{N_{1}}\right]$ (times/d)	[17,191, 17,580]	[17,190, 19,580]
$\left[\underline{N_{e}},\bar{N_{e}}\right]$ (10^8^ times)	[1.3539, 1.6635]	[1.3539, 1.6634]
$\left[\underline{R_{a1}},\bar{R_{a1}}\right]$ (mm)	[0.1868, 0.2077]	[0.1869, 0.2076]

**Table 4 materials-14-02116-t004:** Two kinds of synthetic weight calculation schemes.

Scheme 1				Scheme 2			
Variable $N_{1}$		Variable γ		Variable $N_{1}$		Variable γ	
Subjective weight	Objective weight	Subjective weight	Objective weight	Subjective weight	Objective weight	Subjective weight	Objective weight
$w^{\prime }=0.65$	$w^{\prime \prime }=0.48$	$w^{\prime }=0.35$	$w^{\prime \prime }=0.52$	$w^{\prime }=0.18$	$w^{\prime \prime }=0.66$	$w^{\prime }=0.82$	$w^{\prime \prime }=0.34$
Degree of trust		Degree of trust		Degree of trust		Degree of trust	
a = 0.7	B = 0.3	a = 0.7	B = 0.3	a = 0.55	B = 0.45	a = 0.55	B = 0.45
Synthetic weight		Synthetic weight		Synthetic weight		Synthetic weight	
$q_{N_{1}}=0.6$		$q_{\gamma }=0.4$		$q_{N_{1}}=0.4$		$q_{\gamma }=0.6$	

**Table 5 materials-14-02116-t005:** Equivalent design cumulative axis interval results of different calculation methods.

Calculation Method	Point-Based Algorithm	Scheme 1	**Scheme 2**
$\left[\underline{N_{e}},\bar{N_{e}}\right]$ (10^8^ times)	[1.3539, 1.6634]	[1.3826, 1.6325]	[1.3708, 1.6480]

**Table 6 materials-14-02116-t006:** Interval results of the permanent deformation of asphalt mixture with different calculation methods.

Calculation Method	INTLAB Code Calculation	Point-Based Algorithm	Weight Distribution Based on Interval Variables	
			Scheme 1	Scheme 2
$\left[\underline{R_{a1}},\bar{R_{a1}}\right]$ (mm)	[0.1868, 0.2077]	[0.1869, 0.2076]	[0.1888, 0.2058]	[0.1880, 0.2067]

## Data Availability

All data reported in this paper are contained within the manuscript.

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
