# Peer review of "Calculation Method of Permanent Deformation of Asphalt Mixture Based on Interval Number"

_materials, 2021, doi:10.3390/ma14092116_

Round 1
Reviewer 1 Report
1) Table 2, 4, 5 and 6- please change the font to Palatino.
2) Could your calculations be affected by a measurement error? Because so far no such changes as presented in your article have been made.
3) Could the current algorithmic correlations be fitted to roads not originating from China?
4) In the article, I did not note the laboratory approach confirming the belief that your new approach is fully proven. Why?
5) Could the new calculation method be adapted to asphalts, which are produced from waste materials? This is new trends for roads.
6) Could you explain the reason for changing the existing theoretical model to the one presented in your article? Does the "old" model contain such large deviations?
Author Response
Dear Editor,
Thank you for your enthusiastic work. The following is a detailed reply to the reviewer's comments.
According to the comments of reviewers, I have used the revision mark format in the newly uploaded document, and you will easily see the traces of my revisions based on the review comments.
Yours sincerely,
Yue Xiao
Reviewer #1
Point 1: Table 2, 4, 5 and 6-please change the font to Palatino
Response 1: Thank you for your careful review and valuable comments.
The fonts of Tables 2, 4, 5 and 6 have been changed to Palatino in the newly uploaded manuscript.
Point 2:Could your calculations be affected by a measurement error? Because so far no such changes as presented in your article have been made.
Response 2: This manuscript mainly analyzes the influence of measurement uncertainty on the calculation results, and measurement uncertainty is essentially another type of expression term for measurement error. Measurement uncertainty theory is an error theory developed and gradually improved in the 21st century, and has gradually replaced the application of traditional error theory in many industries.
Point 3:Could the current algorithmic correlations be fitted to roads not originating from China?
Response 3: Of course, this manuscript takes the theoretical model calculation of Chinese highways as an example, mainly introduces the measurement uncertainty theory and proposes an analysis method of interval simplification. This idea and algorithm can be applied to the theoretical analysis of non-Chinese highways.
Point 4:In the article, I did not note the laboratory approach confirming the belief that your new approach is fully proven. Why?
Response 4: Based on the rutting test of asphalt mixture, this manuscript quantifies the measurement uncertainty of the test results during the test process by introducing the theory of measurement uncertainty. Combining interval calculation theory, a simplified calculation of interval theory in road engineering is proposed. The manuscript lists two simplified calculation procedures, which fully proves the effectiveness and feasibility of the new method.
Point 5:Could the new calculation method be adapted to asphalts, which are produced from waste materials? This is new trends for roads
Response 5: The research progress of this manuscript is temporarily unable to prove that the new calculation method can be used for asphalt from waste materials, but the researcher will consider this idea.
Point 6:Could you explain the reason for changing the existing theoretical model to the one presented in your article? Does the "old"model contain such large deviations?
Response 6: (1) The measurement uncertainty theory has been widely used in precision machinery, chemistry, medicine and other fields, and has been recognized by researchers in related industries. However, the material tests in road engineering are mostly rough, and no one realizes the influence of the measurement uncertainty on the final theoretical model calculation results. In particular, after many small measurement uncertainties are gradually accumulated and transmitted, their influence on the analysis of the theoretical model calculation results is even more uninterested. Therefore, it is necessary to analyze the theoretical model of this manuscript.
(2) Not all models have large deviations and should be analyzed in specific models. However, regardless of the model, there is a non-negligible influence of measurement uncertainty, and the researcher should analyze it according to the actual application.

Reviewer 2 Report
Equation 11 is a form of the equation used to predict rutting, why are some values used as intervals while others are deterministics? For example, the temperature, Tpef, is as variable as the axle load frequency interval. Please expand on the choice of when to use intervals and when not to.
Table 4, it is unclear how the weights are assigned. Are they related to the level of trust and is that related to the reliability of the parameter. Please explain.
Conclusion 2, why the need to simplify software? once the code is written and verified, running it is not too critical using modern computers
Overall, this is a good example of the process but needs more explanations.
Author Response
Dear Editor,
Thank you for your enthusiastic work. The following is a detailed reply to the reviewer's comments.
According to the comments of reviewers, I have used the revision mark format in the newly uploaded document, and you will easily see the traces of my revisions based on the review comments.
Yours sincerely,
Yue Xiao
Point 1: Equation 11 is a form of the equation used to predict rutting, why are some values used as intervals while others are deterministics? For example, the temperature, Tpef, is as variable as the axle load frequency interval. Please expand on the choice of when to use intervals and when not to.
Response 1: Thank you for your careful review and valuable comments.
Converting point parameters into intervals mainly considers the uncertainty introduced by the parameter acquisition process and the influence of the parameters on the calculation results. In order to ensure the practicability of introducing measurement uncertainty and interval theory, the theoretical formula must be combined with the practical application of engineering. The uncertainty introduced by parameters such as T_pef in Equation 11 is very small and can be ignored.
Point 2:Table 4, it is unclear how the weights are assigned. Are they related to the level of trust and is that related to the reliability of the parameter. Please explain.
Response 2: Table 4 uses the combined weighting method to calculate the weights of variables, which concentrates the advantages of subjective and objective weighting methods. The calculation method has been described in lines 361-365 of the manuscript and related documents have been cited.
Point 3:Conclusion 2, why the need to simplify software? once the code is written and verified, running it is not too critical using modern computers
Response 3: This manuscript discusses the basic theoretical knowledge in road engineering, but if the point value parameter is converted into an interval, the calculation of the theory will be calculated with the Intlab program. There is an interval expansion phenomenon in the interval calculation result, which will cause the theoretical calculation result to be inconsistent with the application of the actual project, and the limit value of the expansion interval may be different for different physical projects. At this time, the manager needs to control and analyze the expansion result.
Secondly, the measurement uncertainty theory has been widely used in the field of materials and on-site testing in this industry, but the application and analysis of its results are lacking. The main reason is that the theoretical knowledge of interval theoretical calculations has not been popularized. In particular, it must To popularize the theoretical knowledge for the vast number of basic-level personnel in experimental testing.

Round 2
Reviewer 1 Report
The authors meticulously answered the questions I asked. The answers are substantive and I accept them. The article is suitable for further processing as it stands.